# Cardiac Parasympathetic Withdrawal and Sympathetic Activity: Effect of Heat Exposure on Heart Rate Variability

**DOI:** 10.3390/ijerph18115934

**Published:** 2021-05-31

**Authors:** Oriol Abellán-Aynés, Pedro Manonelles, Fernando Alacid

**Affiliations:** 1Faculty of Sport, Catholic University of Murcia (UCAM), 30107 Guadalupe de Maciascoque, Spain; 2International Chair of Sports Medicine, Catholic University of Murcia (UCAM), 30107 Guadalupe de Maciascoque, Spain; pmanonelles@ucam.edu; 3Health Research Center, Department of Education, University of Almeria, 04120 La Cañada, Spain; falacid@ual.es

**Keywords:** sympathetic, parasympathetic, heat, cold, autonomic

## Abstract

*Background*: Research on heart rate variability has increased in recent years and the temperature has not been controlled in some studies assessing repeated measurements. This study aimed to analyze how heart rate variability may change based on environmental temperature during measurement depending on parasympathetic and sympathetic activity variations. *Methods*: A total of 22 volunteers participated in this study divided into an experimental (*n* = 12) and control group (*n* = 10). Each participant was assessed randomly under two different environmental conditions for the experimental group (19 °C and 35 °C) and two identical environmental conditions for the control group (19 °C). During the procedure, heart rate variability measurements were carried out for 10 min. *Results*: Significantly changes were observed for time and frequency domains as well as Poincaré plot variables after heat exposure (*p* < 0.05). These findings were not observed in the control group, whose conditions between measurements did not change. *Conclusions*: The reduction of heart rate variability due to exposure to hot conditions appears to be produced mostly by a parasympathetic withdrawal rather than a sympathetic activation. Therefore, if consecutive measurements have to be carried out, these should always be done under the same temperature conditions.

## 1. Introduction

Heart rate variability (HRV) was initially used merely for clinical purposes [1]. However, in recent years, there has been a growing interest in HRV research and its use. HRV has generally been used for autonomic nervous system assessment [2,3,4], myocardial infarction related assessment [5,6,7], neuropathy associated with diabetes mellitus [8,9], myocardial dysfunction [10,11], and trait anxiety [12], among others. Researchers have suggested that exercise was able to modify autonomic balance [13,14], and at present, HRV is being used even for training control [15,16] and performance estimation [17,18].

Not only does HRV decrease with age, but several variables may also influence changes in HRV. The standard deviation of RR intervals (SDNN), the square root of the mean squared differences of successive RR (RMSSD), and the proportion of interval differences of successive RR intervals greater than 50 ms (pNN50) decrease with age [19]. This is similar for low- and high-frequency power [20] and Poincaré plot variables, such as the standard deviation of instantaneous beat-to-beat RR interval variability (SD1) [21]. Gender is presented as another variable that influences HRV, observing significantly lower values for SDNN and standard deviation of consecutive 5 min RR values in women under 50 years of age [19]. Men show significantly higher values in RMSSD and pNN50 when compared to women at the ages of 29 and above [20], and having no significance in gender from the age range of 50 to 99 years old. Physical fitness appears as another determinant of HRV, observing that subjects with lower values of maximal oxygen consumption also present lower values of high-frequency power and SD1 [21]. Physically active people also present higher values for total power and lower values for high frequency (HF) indexes [22]. Recent findings have been described to assess autonomic balance even more accurately [23].

It is known that hotter environments increase mortality and morbidity if compared to normal conditions. Syncope and cardiovascular collapse may occur as a consequence of affected central hypovolemia by heat stress, which may also endanger tissues as a lack of oxygenation is provoked by changes in blood flow [24]. The main mechanisms for this heat dissipation in humans are supported by increased skin and skeletal muscle blood flow and sweat secretion [25].

Even though several studies have carried out HRV measurement, many of them have not reported if these measurements were taken under the same temperature conditions, considering that the heat and cold stress directly affect many physiological variables [26]. On the topic of whether it is important to take into account the temperature for the sympathetic-vagal balance assessment, some previous studies have described how temperature affects HRV depending on the ambient temperature of the season but not during the measurement [27]. Additionally, some articles have studied the effect of different temperatures on HRV during the measurement [28,29,30], but not all of them have analyzed time domain, frequency domain, and Poincaré plot variables. Thus, this study aimed to analyze how heart rate variability may change based on the environmental temperature during measurements, depending on the parasympathetic and sympathetic variations.

## 2. Materials and Methods

A total of 22 physically active (11 men and 11 women), healthy, and non-smoker individuals were randomly divided into an experimental group (EG) and control group (CG). Table 1 shows the characteristics of both groups. All subjects voluntarily participated in this study and signed an informed consent form before the beginning of the measurements. The study was conducted according to the guidelines of the Declaration of Helsinki and approved by the ethics committee of the Catholic University of Murcia (reference 6714).

Each participant attended the laboratory for two consecutive days at the same time of the day (15:00). During the first visit, anthropometric measurements were assessed (height and weight) and the first heart rate variability measurement was performed immediately after anthropometric measurements. Body mass and stretch stature were assessed using a scale with stadiometer Seca 720 (Hamburg, Germany) with a precision of 100 g and 1 mm, respectively. The succeeding day, participants went to the laboratory again for the second heart rate variability measurement. All participants were requested to avoid any high-intensity activity, to avoid eating copiously and refrain from alcohol and coffee consumption 24 h before the first measurement until the end of the second measurement. Additionally, none of the women were measured during the menstrual period.

HRV measurements consisted of 10 min in a lying down position under two different environmental conditions, hot environment (HE) at 35 °C and cool environment (CE) at 19 °C for EG and two exposures to a 19 °C environment for the CG. The temperature selected for each measurement was reached before each participant came to the laboratory. Relative humidity during the measurements remained constant to avoid it as an external factor affecting the HRV measurements. The relative humidity constant value was 51%. Ambient temperature and relative humidity were assessed before and during the HRV measurements using an environmental meter Kestrel 5000 (Minneapolis, MN, USA). The temperature variations were in a range of ±0.4 °C and the relative humidity was in a range of ±0.3%. Individuals wore shorts and a sports shirt and did not wear any shoes or socks. The order of each trial was randomly assigned. Subjects wore a heart rate sensor Polar H7 (Kempele, Finland) to assess beat-to-beat recordings during the trial. Participants rested in lying position 10 min before the beginning of the test to ensure that resting HR was reached.

Further variable calculation was undertaken using the software Kubios HRV version 3.0. This software was also used to apply low threshold filters to remove artifacts; only when artifacts appeared, a filter was used. Time domain, frequency domain, and Poincare plot variables were retrieved. An analysis of data was done in a 5-min interval and two different HRV analyses were assessed for each person and trial. Measuring was done from 0 to 5 min and from 5 to 10 min, as recommended [31]. The variables analyzed were the mean R-R interval time (RR), the standard deviation of all RR intervals (SDNN), the square root of the mean squared differences of successive RR intervals (RMSSD), and the number of interval differences of successive RR intervals greater than 50 ms (NN50) and its proportion (pNN50) for time domain variables. Fast Fourier Transform (FFT) was used to calculate the spectral components of the frequency domain. Low-frequency power (0.04–0.15 Hz), high-frequency power (0.15–0.4 Hz), and total power (0.04–0.4 Hz) components were calculated as integrals of the respective power spectral density curve. These variables were expressed in natural logarithm transformed values (LFln, HFln, and TPln). Simultaneously, the LF/HF ratio was calculated, and LF and HF were expressed in normalized units to study the balanced behavior of sympathetic and parasympathetic nervous systems (LFnu, HFnu). Geometric pattern variables as the HRV triangular index and the triangular interpolation of RR interval histogram and very low-frequency power were not calculated due to the short time of the HRV measurement. Furthermore, the Poincaré plot-derived variables, such as the standard deviation of the instantaneous beat-to-beat RR interval variability (SD1), the standard deviation of continuous long-term R-R interval variability (SD2), and the SD2/SD1 ratio, were calculated. Moreover, to assess the cardiac sympathetic tone, we calculated the stress score (SS), which is calculated through “SS = 1000 ∗ 1/SD2” [23] as the SD2 behavior has been shown to be inverse to the sympathetic activity [23]. Finally, to assess the sympathetic-vagal balance, we calculated the sympathetic/parasympathetic ratio (S/PS) through the relation between SD1 (parasympathetic tone) and SS (sympathetic tone) [23].

Statistical analyses were conducted with IBM statistics SPSS v20.0 software (SPSS Inc., Chicago, IL, USA). A Shapiro–Wilk test was used for testing the normality of the data and a two-way analysis of variance was used to investigate the main effects and the interaction between the group factor (experimental vs. control) and the temperature factor (19 °C vs. 35 °C) as well as for the interaction between the group and temperature factor. A Bonferroni post-hoc test was performed and the eta squared partial (*η*^2^*_p_*) was calculated to assess the differences and effect size of the comparisons. A level of *p* ≤ 0.05 was set to indicate statistical significance and effect sizes were set as small; *η*^2^*_p_* < 0.13, medium; *η*^2^*_p_* < 0.26, and large; *η*^2^*_p_* > 0.26. Results are expressed as mean values ± standard deviation.

## 3. Results

We did not observe any difference between the first and last 5 min of measurements in any of the variables assessed during the 19 °C exposure (*p* > 0.05). No significant differences were found between EG and CG (*p* > 0.05) for any HRV variable, thus demonstrating the homogeneity of the sample in both groups.

Table 2 and Table 3 show the data for the assessment comparing 19 °C and 35 °C during the first and last 5 min, respectively. Both tables show the pair comparisons as well as the ANOVA interactions for the temperature and group effect as well as the interaction Group*temperature effect.

## 4. Discussion

The main findings of this study were that not only do time domain variables significantly change as a consequence of temperature, but the frequency domain and Poincare plot variables are also affected. The ambient temperature effects on HRV were equal between the first and last 5 min of measurement. Moreover, these differences did not appear in the CG after performing both measurements in the same conditions. These variations of HRV were higher in parasympathetic-related variables than in increments of sympathetic-related variables.

Our results were similar to those recently revealed by other researchers, who observed that hotter environments significantly reduced SDNN and RMSSD values [32]. On the contrary, we did not observe the differences between CE and HE in SDNN values as shown in previous research [28], in which an absence of significant differences on SDNN can be seen when comparing a normal environment to a HE, observing significantly lower RR intervals during HE. Additionally, pNN50 was always lower during heat exposure. The standards of measurement, physiological interpretation, and clinical use [31] recommended not to use NN50 and pNN50 because of their low statistical properties. Instead of these variables, it is recommended to use RMSSD, which also decreases as a consequence of heat exposure. However, low values of pNN50 have typically been used as a prognostic indicator of chronic heart failure and myocardial infarction, as well as SDNN [5,33,34]. The RMSSD is the most commonly used variable in HRV to assess the autonomic nervous system activity, and our results have demonstrated that higher temperatures induce a greater parasympathetic withdrawal during resting position. This phenomenon shows how higher temperatures may behave as an external event which may affect real values if several measurements have to be undertaken during the time. Further research should study this effect not only during resting position but also during exercise and post-exercise recovery.

Other studies determined that under different temperature conditions, LFnu showed higher values during hot conditions, but similar results were not present when comparing normal and cold conditions [29,30]. Conversely, HFnu presented lower values during HE, revealing no differences between cold and normal conditions [29,30]. These results were in agreement with our findings concerning LFnu, showing an increment during heat exposure. Our results also showed decreases in HFnu due to higher temperature exposure. These changes in LFnu and HFnu indicate that the proportion of the sympathetic-vagal activation is altered by high temperatures [5,35]. We have already mentioned that HF has a relationship with parasympathetic activity; however, LF has no relationship with sympathetic activity, and therefore, we do not interpret LFnu as a variable of sympathetic activity. On the other hand, LFnu can be interpreted as the inverse of parasympathetic activity, since a higher value of HFnu will result in a lower value of LFnu. Significantly lower values were observed in absolute values instead of natural logarithm transformed values for TP, very low frequency (VLF), LF, and HF comparing hot and normal conditions [28,30]. Although significantly higher values were described for LF/HF in a hotter environment in a previous study [30], the effect of temperature on LF/HF was not detected in our data. Moreover, significantly higher values were found for HFln during heat exposure; these differences did not appear in the CG. We did not assess VLF due to the low validity in short-term measurements [31]. Heat exposure provoked reductions of HFln, which means that the frequency domain balance is altered by temperature. Since HRV variables have commonly been used for training load and control, with HF being more interesting [15,16,17,18,36], athletes and coaches should take into account that each HRV assessment must be undertaken in the same temperature conditions. This could be a challenge for individual or team sports, where the athlete’s training load is controlled by HRV, and competing in different environments with varying hot and cold temperatures. In these cases, environmental variables may produce misleading data.

Previous research showed that after parasympatholytic infusion, SD1 and SD2 started to decrease, and it approached closer to zero the more atropine was infused [37], showing the direct relationship between SD1 and parasympathetic activation. Since heat exposure reduces SD1 values, measuring HRV in changing conditions would make HRV analysis ineffective for the control of vagal modulation of HR; this effect does not require SD1 calculation—calculating RMSDD is enough, since both variables are identical [38]. A previous investigation revealed that after atropine infusion and during exercise, SD2/SD1 values increased similarly to those reported in the present study under heat exposure [37]. In previous research, the SD2/SD1 ratio was shown to not be as clear as an indicator of autonomic balance [23]; our results agree with that because it was not stable from CE5 to HE5. To avoid this lack of stabilization, the SS and S/PS ratio were previously defined [23], showing that the inverse of SD2 has a direct relationship to sympathetic activity, and thus, the ratio S/PS (SS/SD1) may give us information about the overlap of sympathetic activation over parasympathetic in our measurements. SS did not change due to heat exposure, showing that sympathetic activity is not significantly increased. These new indexes show a greater relationship with autonomic balance variables than the SD2/SD1 ratio. This means that higher temperature produces a sympathetic activation assessed through Poincare plot variables of HRV.

The main interest in using lesser-known variables such as SS is because they have been created specifically to represent sympathetic activity. While it is true that other HRV-related variables have traditionally been used to control sympathetic activity such as LF or the LF/HF ratio, in recent years, a low relationship of sympathetic activity and these variables has been demonstrated. This is why a new perspective has been opened up to quantify the sympathetic variant using variables created specifically for this purpose, such as SS, and the ratio between SS and the already known variables for quantifying the parasympathetic side, such as SD1 or the identical RMSSD.

An interesting fact observed in the present work is that during the second 5-min section of heat exposure, HRV is reduced with respect to the first section. This seems to indicate that longer heat exposure may still cause greater parasympathetic depletion. It might be interesting for future research to analyze how HRV varies as a function of heat exposure time and whether a stable plateau is reached as a function of different temperatures.

Considering that the sample in this study was not acclimatized to heat, the influence of temperature on the organism must have triggered an excitability of the ANS, causing an increase in blood flow that would subsequently translate into a significant increase in HR. These changes in HR, together with the buffering of heat by the ANS through blood flow, appear to be responsible for the significant changes observed in HRV with heat exposure.

The data analyzed in this study are coherent as increments in heart rate during heat exposure are related to a decrease in parasympathetic effects [39,40] and were previously described to be as a consequence of 25% of sympathetic activation and 75% of withdrawal of parasympathetic activity [41], showing that heat affects more markedly the parasympathetic component than the sympathetic component.

The strength of the current research is that an important external factor has been defined when consecutive HRV measurements have to be carried out, defining how ambient temperature can affect the assessment of an easy and cheap analysis method. Additionally, the control of cardiac parasympathetic activity can be quantified simply with a heart-rate band. Otherwise, some limitations are present, such as the sample size or the specificity of the sample, which must be taken into account when interpreting the results.

## 5. Conclusions

Since the use of HRV has become more popular nowadays for several purposes, such as clinical use, training workload assessment, or sympathetic-vagal modulation, it is important to take into account what kind of variables might influence HRV measurements. The presented findings and investigation show great changes in HRV variables, suggesting that different temperature conditions influence HRV parameters of time, frequency domain, and Poincare plot variables. These finding are strongly coherent due to the heat as a vasodilator. Thus, HRV measurements must always be carried out under the same temperature conditions for the same person if several measurements have to be taken; otherwise, the results might be altered. Moreover, the reduction of heart rate variability due to the exposure to hot conditions appears to be produced mostly by a parasympathetic withdrawal rather than a sympathetic activation.

## Figures and Tables

**Table 1 ijerph-18-05934-t001:** Sample characteristics.

	Experimental Group (*n* = 12)		Control Group (*n* = 10)	
Outcome	M	SD	M	S
Age (years)	25	3	24	3
Height (cm)	174	7	176	8
Weight (kg)	68	11	71	9
Body mass index (kg∙m^2^)	22.4	2.2	22.9	1.4

**Table 2 ijerph-18-05934-t002:** Differences on heart rate variability depending on ambient temperature in experimental and control groups during first 5 minutes of measurement.

							ANOVA (*F*, *p*, *η*^2^*_p_*)								
		19 °C		35 °C EG 19 °C CG			Temperature Effect			Group Effect			Group*Temperature Effect		
Outcome	Group	M	SD	M	SD	*p*	*F*	*p*	*η* ^2^ *_p_*	*F*	*p*	*η* ^2^ *_p_*	*F*	*p*	*η* ^2^ *_p_*
RR	EG	1066.44	184.00	876.17	326.35	0.004	13.157	0.002	0.409	0.015	0.905	0.001	3.848	0.065	0.168
	CG	982.28	177.69	996.61	159.77	1									
SDNN	EG	75.43	21.80	64.25	20.73	0.333	0.025	0.876	0.001	1.224	0.282	0.061	2.34	0.143	0.11
	CG	71.43	21.13	73.90	21.54	0.258									
RMSSD	EG	76.14	28.92	52.85	29.46	0.096	0.38	0.545	0.2	0.087	0.771	0.005	3.255	0.087	0.146
	CG	60.74	22.37	61.18	26.74	0.422									
pNN50	EG	45.74	18.90	28.43	22.26	0.021	2.338	0.143	0.11	0.612	0.444	0.031	3.72	0.069	0.164
	CG	38.76	17.20	37.35	27.75	0.785									
LF_ln_ (ms^2^)	EG	7.14	0.66	6.95	0.70	0.449	0.051	0.824	0.003	0.06	0.81	0.003	1.671	0.212	0.081
	CG	6.89	0.77	6.85	0.84	0.307									
LF_nu_	EG	44.35	21.43	52.60	18.58	0.049	0.7	0.413	0.036	1.154	0.296	0.057	3.593	0.073	0.159
	CG	56.49	17.60	50.97	21.68	0.473									
HF_ln_ (ms^2^)	EG	7.4	0.75	6.82	0.89	0.032	0.179	0.677	0.009	1.495	0.236	0.073	4.034	0.059	0.175
	CG	6.67	0.89	6.72	1.38	0.287									
HF_nu_	EG	55.63	21.44	47.20	18.36	0.049	0.782	0.388	0.04	1.172	0.293	0.058	3.68	0.07	0.162
	CG	43.38	17.60	48.90	21.69	0.484									
TP_ln_	EG	8.38	0.50	8.06	0.61	0.233	0.001	0.98	<0.001	0.007	0.932	<0.001	2.981	0.1	0.136
	CG	6.64	1.79	6.51	1.55	0.241									
LF/HF	EG	1.30	1.72	1.68	1.95	0.281	0.985	0.333	0.049	0.214	0.649	0.011	0.292	0.595	0.015
	CG	1.94	0.59	1.53	0.57	0.758									
SD1	EG	53.95	20.48	40.20	18.62	0.095	0.380	0.545	0.02	0.086	0.773	0.004	3.268	0.087	0.147
	CG	43.04	15.85	49.00	37.37	0.421									
SD2	EG	91.08	26.65	81.07	24.30	0.491	0.15	0.703	0.008	1.665	0.212	0.081	1.842	0.191	0.088
	CG	91.46	29.72	89.93	22.93	0.243									
SD2/SD1	EG	1.82	0.58	2.19	0.59	0.04	3.153	0.092	0.142	1.503	0.235	0.073	1.597	0.222	0.078
	CG	2.96	2.09	2.21	0.69	0.728									
SS	EG	10.97	2.62	13.32	3.83	0.187	0.511	0.483	0.026	0.91	0.352	0.046	1.375	0.255	0.067
	CG	13.28	3.34	12.92	3.81	0.755									
S/PS	EG	0.23	0.12	0.43	0.27	0.045	1.852	0.189	0.089	0.052	0.822	0.003	2.566	0.126	0.119
	CG	0.32	0.30	0.50	0.42	0.87									

HR: Heart rate; SDNN: Standard deviation of consecutive R-R intervals; pNN50: relative value of consecutive intervals that differ by more than 50 ms; RMSSD: root mean square of successive differences of consecutive R-R intervals; HF_ln_: High-frequency power based on its natural logarithm; LF_ln_: Low-frequency power based on its natural logarithm; HF_nu_: High-frequency power in normalized units; HF_nu_: High-frequency power in normalized units; LF/HF: Ratio between low-and high-frequency power; SD1: the standard deviation of instantaneous beat-to-beat interval variability; SD2: continuous long-term R/R interval variability; SS: Stress Score; S/PS: sympathetic-parasympathetic ratio.

**Table 3 ijerph-18-05934-t003:** Differences on heart rate variability depending on ambient temperature in experimental an control groups during last 5 minutes of measurement.

							ANOVA (*F*, *p*, *η*^2^*_p_*)								
		19 °C		35 °C EG 19 °C CG			Temperature Effect			Group Effect			Group*Temperature Effect		
Outcome	Group	M	SD	M	SD	*p*	*F*	*p*	*η* ^2^ *_p_*	*F*	*p*	*η* ^2^ *_p_*	*F*	*p*	*η* ^2^ *_p_*
RR	EG	1078.84	174.80	932.56	128.32	<0.001	17.093	0.001	0.474	0.08	0.78	0.004	4.127	0.056	0.178
	CG	1018.06	127.96	1048.88	112.37	0.163									
SDNN	EG	76.42	21.54	65.52	20.47	0.932	0.348	0.562	0.018	2.383	0.139	0.111	0.503	0.487	0.026
	CG	71.83	21.67	74.02	22.32	0.381									
RMSSD	EG	79.38	32.51	52.37	26.70	0.042	0.58	0.456	0.03	0.123	0.73	0.006	3.62	0.072	0.16
	CG	61.50	33.54	61.45	29.44	0.44									
pNN50	EG	46.72	17.64	27.64	18.07	0.019	3.395	0.081	0.152	0.138	0.714	0.007	2.882	0.106	0.132
	CG	39.77	23.16	37.57	20.81	0.921									
LF_ln_ (ms^2^)	EG	7.08	0.74	6.95	0.63	0.655	0.546	0.469	0.028	0.233	0.635	0.012	0.013	0.912	0.001
	CG	6.84	0.79	6.84	0.64	0.563									
LF_nu_	EG	41.70	18.97	57.05	19.47	0.026	1.876	0.187	0.09	0.041	0.842	0.002	3.821	0.065	0.167
	CG	58.25	15.95	51.99	18.03	0.691									
HF_ln_ (ms^2^)	EG	7.44	0.75	6.64	0.89	0.038	1.86	0.189	0.089	0.422	0.524	0.022	1.888	0.185	0.09
	CG	6.47	1.15	6.72	1.03	0.994									
HF_nu_	EG	58.22	18.98	42.89	19.46	0.026	1.865	0.188	0.089	0.044	0.836	0.002	3.841	0.065	0.168
	CG	41.65	15.85	47.84	18.03	0.686									
TP_ln_	EG	8.51	0.55	8.16	0.66	0.156	1.903	0.184	0.091	0.42	0.525	0.022	0.434	0.518	0.022
	CG	6.65	1.03	6.78	0.84	0.624									
LF/HF	EG	1.04	1.24	1.82	1.31	0.47	0.5	0.488	0.026	<0.001	0.988	<0.001	2.505	0.13	0.116
	CG	1.77	0.36	1.57	0.50	0.493									
SD1	EG	56.24	23.04	37.08	18.92	0.042	0.583	0.455	0.03	0.124	0.729	0.006	3.625	0.072	0.16
	CG	43.56	23.76	43.53	20.83	0.44									
SD2	EG	91.66	23.08	83.95	25.63	0.473	0.211	0.651	0.011	0.203	0.657	0.011	0.303	0.588	0.016
	CG	90.92	31.15	87.92	24.84	0.95									
SD2/SD1	EG	1.75	0.47	2.54	0.79	0.048	2.924	0.104	0.133	0.034	0.856	0.002	5.062	0.037	0.21
	CG	2.29	0.49	2.23	0.61	0.238									
SS	EG	13.32	3.83	12.89	3.65	0.337	1.311	0.266	0.065	0.002	0.962	<0.001	0.046	0.833	0.002
	CG	13.65	3.31	12.72	3.36	0.528									
S/PS	EG	0.25	0.15	0.47	0.35	0.041	3.681	0.07	0.162	0.257	0.618	0.013	0.463	0.504	0.024
	CG	0.38	0.07	0.37	0.07	0.403									

HR: Heart rate; SDNN: Standard deviation of consecutive R-R intervals; pNN50: relative value of consecutive intervals that differ by more than 50 ms; RMSSD: root mean square of successive differences of consecutive R-R intervals; HF_ln_: High-frequency power based on its natural logarithm; LF_ln_: Low-frequency power based on its natural logarithm; HF_nu_: High-frequency power in normalized units; HF_nu_: High-frequency power in normalized units; LF/HF: Ratio between low-and high-frequency power; SD1: the standard deviation of instantaneous beat-to-beat interval variability; SD2: continuous long-term R/R interval variability; SS: Stress Score; S/PS: sympathetic-parasympathetic ratio.

## Data Availability

The data presented in this study are available on request from the corresponding author.

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
