# Peer review of "Cardiac Parasympathetic Withdrawal and Sympathetic Activity: Effect of Heat Exposure on Heart Rate Variability"

_ijerph, 2021, doi:10.3390/ijerph18115934_

Round 1

Reviewer 1 Report

Although HRV measurement is a "popular method", it is a work that does not contribute sufficiently to the knowledge, its practical application and its projection is limited. Although it has a coherent method, it lacks some important aspects.

It is not described how the participants were assigned to each group (EG and CG) (line 70).

The approval number of the scientific ethics committee is not described (line 74).

It is not described how many women and men participated in the study (line 84).

The “Poincaré plot” is exposed in the methodology section, and also in the conclusions, but it is not appreciated in the results. This is not clear (line 111).

The results of the t-student test are not shown in Tables 1 and 2. The t-student test cannot replace ANOVA. This is not clarified (line 119).

ANOVA is applied Why was a post hoc test not used? (line 120 and table 1 “RR outcome”).

The ranges to classify the effect size are not indicated: example: low, moderate, high (line 122).

A table is necessary with the characteristics of weight, height, age, BMI… of both groups (line 126).

The article does not present strengths and limitations (line 214).

It does not present the authors' contributions section.

Thank you.

Author Response

Reviewer 1.

Although HRV measurement is a "popular method", it is a work that does not contribute sufficiently to the knowledge, its practical application and its projection is limited. Although it has a coherent method, it lacks some important aspects.

Firslty, we would like to thank reviewer #1 for the advice and details to improve our work, for being clear and concise, and for helping us move forward in the review process.

It is not described how the participants were assigned to each group (EG and CG) (line 70).

Participants were assigned to each group randomly. This information has been added in the text.

The approval number of the scientific ethics committee is not described (line 74).

The reference number has been added

It is not described how many women and men participated in the study (line 84).

The information about how many men and women participated in the study is now shown in line 69.

The “Poincaré plot” is exposed in the methodology section, and also in the conclusions, but it is not appreciated in the results. This is not clear (line 111).

In the methods section (112-117) it is explained that Poincaré Plot variables are SD1 and SD2 as well as the variables derived from SD1 and SD2. In the results section SD1, SD2, SD2/SD1, SS and S/PS are shown.

The results of the t-student test are not shown in Tables 1 and 2. The t-student test cannot replace ANOVA. This is not clarified (line 119). ANOVA is applied Why was a post hoc test not used? (line 120 and table 1 “RR outcome”).

We made a mistake writing the methods section. We did not use the t-student test, we used a Bonferroni post-hoc analysis after the ANOVA.

The ranges to classify the effect size are not indicated: example: low, moderate, high (line 122).

We have indicated the classification of the effect sizes.

A table is necessary with the characteristics of weight, height, age, BMI… of both groups (line 126).

We have added the table 1 with the information of the sample and we have also deleted this information from the text.

The article does not present strengths and limitations (line 214).

In line 217 we have added this part.

It does not present the authors' contributions section.

We have presented this section but the journal asked to attach it independently from the text. The section is as follows:

Conceptualization, Oriol Abellán-Aynés; Data curation, Oriol Abellán-Aynés; Formal analysis, Oriol Abellán-Aynés; Investigation, Oriol Abellán-Aynés; Supervision, Pedro Manonelles and Fernando Alacid; Writing – original draft, Oriol Abellán-Aynés.

Thank you.

Reviewer 2 Report

The paper from Abellán-Aynés is a simple but interesting study that evaluates the effect of the environmental temperature on HRV. The general writing could be improved, especially the Introduction and the Discussion.

General Comments

It would be nice to report the body temperature and the arterial pressure of the participants in the different conditions studied. Did you collect those variables?

The authors argued that increased temperature is related to parasympathetic withdrawal rather than sympathetic activation. However, in my opinion, results point to both decreased parasympathetic and increased sympathetic control. In both 5-min segments, LF nu is increased, HF abs is decreased, and S/SP is increased. Assessing LF nu and HF abs (instead of both in normalized units) seems to be a better strategy since LFnu + HFnu = 100, and thus, cannot represent independently the sympathetic and parasympathetic modulation.

Specific comments

In the Abstract and Conclusion, you say that sympathetic control is not altered in HE. However, in the last paragraph of the Discussion, you say the opposite. Please, revise.

Lines 48-49: physical training is supposed to increase vagal modulation. Should not they show a higher HF component of HRV?

Line 57: there are too many studies with HRV to say that none of them have reported the temperature conditions. I agree that it is not usual, but we cannot assure all of them lack this information.

Lines 61-63: I did not understand this sentence.

Lines 69-72: how many men/women in each group?

Line 76: what is 1500? Would it be 3 p.m.?

Lines 96-98: what is the rationale for dividing the 10-min recording into two periods? Also, how did you deal with these two 5-min series for the comparison of the two days? Did you average RHV indices from the two excerpts? Please, clarify.

Lines 114-116: SS and S/PS are not usual HRV indices. Please, elaborate a little further on it (maybe in a new paragraph).

Lines 122-123: how should eta squared partial be interpreted?

Line 129: Comparison between EG and CG were not different for 19º only? We would expect differences when the temperature is not the same. Please, clarify.

Line 163: RMSSD decreases in only one segment of 5-min. This paragraph emphasized the changes observed in RMSSD but the results are not consistent.

Line 199: since SD1 and RMSSD are identical, why did you calculate both?

Lines 221-222: heat acting as a vasodilator and the consequences in the autonomic system deserve a paragraph in the Discussion section. It is an important interpretation and you only mentioned it in the conclusion.

Author Response

Reviewer 2

We would like to thank reviewer 2 for asking us questions that the readers might have and that have helped us to provide more concise explanations in the text to clarify doubts. In addition, he has made a very interesting contribution by proposing that we shold not only focus on addressing HRV variables but also on providing physiological explanations about the results we have obtained.

General Comments

It would be nice to report the body temperature and the arterial pressure of the participants in the different conditions studied. Did you collect those variables?

Unfortunately, we did not measure these variables as we intended to focus on the direct effect of temperature on HRV and this was the only measurement we carried out to draw out the exposed variables.

The authors argued that increased temperature is related to parasympathetic withdrawal rather than sympathetic activation. However, in my opinion, results point to both decreased parasympathetic and increased sympathetic control. In both 5-min segments, LF nu is increased, HF abs is decreased, and S/SP is increased. Assessing LF nu and HF abs (instead of both in normalized units) seems to be a better strategy since LFnu + HFnu = 100, and thus, cannot represent independently the sympathetic and parasympathetic modulation.

The variable we used to assess the sympathetic activity was SS. We chose this one because it is calculated as the inversion of SD2. Previous research observed that SD2 has an inverse behavior to sympathetic activity (Tulppo, 1996), thus creating a new variable from the inverse of SD2 intends to asses sympathetic activity (Orellana, 2015). We agree with reviewer 2 about the use of LF and HF abs to asses parasympathetic activity with HF (it is presented in table 2 and 3 based on ln) but HFnu can give information about how the high-frequency component can change also based on the low-frequency component. Reviewer 2 is totally right mentioning that S/PS is increased, however we have seen in our results that RMSSD and SD1 are significantly decreased but SS does not increase significantly. Thus, this change in S/PS seems to be due to the parasympathetic withdrawal rather than a sympathetic-activity increase

Tulppo, M. P., Makikallio, T. H., Takala, T. E., Seppanen, T. H. H. V., & Huikuri, H. V. (1996). Quantitative beat-to-beat analysis of heart rate dynamics during exercise. American journal of physiology-heart and circulatory physiology271(1), H244-H252.

Orellana, J. N., de la Cruz Torres, B., Cachadiña, E. S., de Hoyo, M., & Cobo, S. D. (2015). Two new indexes for the assessment of autonomic balance in elite soccer players. International journal of sports physiology and performance, 10(4), 452-457.

Specific comments

In the Abstract and Conclusion, you say that sympathetic control is not altered in HE. However, in the last paragraph of the Discussion, you say the opposite. Please, revise.

Lines 48-49: physical training is supposed to increase vagal modulation. Should not they show a higher HF component of HRV?

 Thank you for the observation, the last paragraph of the discussion was a little difficult to understand and we have added one more sentence to make it easier to understand our point

Line 57: there are too many studies with HRV to say that none of them have reported the temperature conditions. I agree that it is not usual, but we cannot assure all of them lack this information.

 It is true that the sentence used is not correct because it is excessive to mention that none has controlled the temperature. We have corrected this sentence because it is incorrect to mention that no study has controlled the ambient temperature.

Lines 61-63: I did not understand this sentence.

 It was a complicated sentence to read since the same word was used several times. We have corrected it to make it easier for readers to understand.

Lines 69-72: how many men/women in each group?

Another reviewer told us exactly the same, it is an issue to correct. We have added the information in line 69.

Line 76: what is 1500? Would it be 3 p.m.?

 Yes, we have now indicated it as 15:00. We prefer to show it as 15:00 but in previous articles the reviewers have asked us to represent it with the format 1500 but we also thing it is difficult to understand. We did it just in case other reviewers think it is more correct the other way

Lines 96-98: what is the rationale for dividing the 10-min recording into two periods? Also, how did you deal with these two 5-min series for the comparison of the two days? Did you average RHV indices from the two excerpts? Please, clarify.

Commonly, in science in general, 5-minute measurements are used. We wanted to reproduce the same type of measurement that is normally performed, in this case analyzing the effect of temperature. That is why we did not want to analyze a single 10-minute measurement. Basically, we performed the 10-minute measurement and with the kubios software we analyzed the two 5-minute sections individually. This was also interesting to us when analyzing the data as we observed a decrease in HRV in the second section. This also helped us to understand that the longer the exposure time, the greater the effect of temperature on HRV. It was also a mistake on our part not to mention this better in the discussion. We have added a paragraph explaining this fact. Lines 219-223

Lines 114-116: SS and S/PS are not usual HRV indices. Please, elaborate a little further on it (maybe in a new paragraph).

It is true that when we are used to working with variables that other people may not know, we tend to take things for granted and do not go into sufficient depth to help readers become familiar with these new data. To facilitate this, we have added a paragraph explaining why we find it more interesting to use variables such as SS and/or S/PS to quantify the sympathetic component. Lines 213-220

Lines 122-123: how should eta squared partial be interpreted?

Another reviewer also noticed us with that issue. We made a mistake writing the statistical analysis section. We have added the information

Line 129: Comparison between EG and CG were not different for 19º only? We would expect differences when the temperature is not the same. Please, clarify.

Since our aim was to observe the direct effect of temperature on HRV in the same group of subjects, we did not make the comparison between groups at different temperatures as this would not be reproducible data. The fact of comparing both groups at the same temperature was done with the aim of establishing the homogeneity of the sample. Therefore, we only focused on observing the effect of temperature change on the group subjected to different temperatures and not on comparing two different groups under different conditions. In line 129 we have explained why to facilitate understanding among readers.

Line 163: RMSSD decreases in only one segment of 5-min. This paragraph emphasized the changes observed in RMSSD but the results are not consistent.

 This is what we explained earlier, that it seems that a plateau is not reached with exposure to heat, since this difference is observed in the RMSSD from minute 5 to 10

Line 199: since SD1 and RMSSD are identical, why did you calculate both?

That's right. The reason for calculating RMSSD is because it is the primary variable of the HRV. On the other hand, the value of SD1 is necessary to calculate the S/PS ratio.

Lines 221-222: heat acting as a vasodilator and the consequences in the autonomic system deserve a paragraph in the Discussion section. It is an important interpretation and you only mentioned it in the conclusion.

It is true. Since this was an article focused on HRV, we have concentrated most of the efforts in the discussion on HRV variables, leaving aside the physiological proposals to explain the observed results. We have added a brief paragraph dealing with the matter in order to be consistent with the conclusions section.

Reviewer 3 Report

I would like to request rationale of conducting a study designfor this research question.Generally, intergroup comparative studies is conducted by accurate calculation of sample size with previous observational studies. In this study, sample size may be relatively small. 
 Fact that ethical committee permitted the study would not be sufficient in this case. Additionally, in this review, scientifically sound sample size calculation could be essential.

Author Response

Reviewer 3

We would like to thank reviewer 3 for his valuable comments to improve the quality of our work.

I would like to request rationale of conducting a study designfor this research question.Generally, intergroup comparative studies is conducted by accurate calculation of sample size with previous observational studies. In this study, sample size may be relatively small. 
 Fact that ethical committee permitted the study would not be sufficient in this case. Additionally, in this review, scientifically sound sample size calculation could be essential.

We understand that calculations of the sample size is essential for this type of research. However, it was only possible to obtain the 22 subjects that form the study sample and we carried out the research with them. We have added a section on the limitations of the study so that readers will have to consider that this fact is important when interpreting the results.

We have added the reference number of the ethics committee approval.

Round 2

Reviewer 1 Report

Dear authors,

As a strength, I suggest including the instrument used to measure HRV (Polar H7 heart rate sensor).

Thank you.

Author Response

We would like to thank reviewer 1 for the work done to improve our article.

As a strength, I suggest including the instrument used to measure HRV (Polar H7 heart rate sensor)

We have taken into consideration the comment on the strengths of the article and have included it.

Thank you very much

Reviewer 2 Report

The variable we used to assess the sympathetic activity was SS. We chose this one because it is calculated as the inversion of SD2. Previous research observed that SD2 has an inverse behavior to sympathetic activity (Tulppo, 1996), thus creating a new variable from the inverse of SD2 intends to asses sympathetic activity (Orellana, 2015). We agree with reviewer 2 about the use of LF and HF abs to asses parasympathetic activity with HF (it is presented in table 2 and 3 based on ln) but HFnu can give information about how the high-frequency component can change also based on the low-frequency component. Reviewer 2 is totally right mentioning that S/PS is increased, however we have seen in our results that RMSSD and SD1 are significantly decreased but SS does not increase significantly. Thus, this change in S/PS seems to be due to the parasympathetic withdrawal rather than a sympathetic-activity increase

I understand your point that SS has been demonstrated to be related to sympathetic modulation. However, this is not a usual index and there are plenty of papers showing the association of LFnu with sympathetic modulation, and LFnu is increased in your sample. If both are associated with sympathetic control, this is an ambiguous finding that does not allow a strong interpretation. Concluding that there is a sympathovagal imbalance, shifted towards the sympathetic branch, seems to be a more reasonable interpretation.

It is true that when we are used to working with variables that other people may not know, we tend to take things for granted and do not go into sufficient depth to help readers become familiar with these new data. To facilitate this, we have added a paragraph explaining why we find it more interesting to use variables such as SS and/or S/PS to quantify the sympathetic component. Lines 213-220

Those lines are not describing SS and S/PS. Moreover, this description should be added to the “Methods” section.

It is true. Since this was an article focused on HRV, we have concentrated most of the efforts in the discussion on HRV variables, leaving aside the physiological proposals to explain the observed results. We have added a brief paragraph dealing with the matter in order to be consistent with the conclusions section.

Lines 229-234: The authors say “…the influence of temperature on the organism must have triggered excitability of the ANS, causing an increase in blood flow… “. By “increase in blood flow” you mean the cardiac output? Is not vasodilation the effect expected from heat increase?

Author Response

I understand your point that SS has been demonstrated to be related to sympathetic modulation. However, this is not a usual index and there are plenty of papers showing the association of LFnu with sympathetic modulation, and LFnu is increased in your sample. If both are associated with sympathetic control, this is an ambiguous finding that does not allow a strong interpretation. Concluding that there is a sympathovagal imbalance, shifted towards the sympathetic branch, seems to be a more reasonable interpretation.

 We understand the point of interpreting LFnu as sympathetic tone. However, since more recent studies confirm that HF does have a relationship with parasympathetic activity and that LF has no relationship with sympathetic activity [Martelli, Silvani, McAllen, May, Ramchandra (2014) The low frequency power of heart rate variability is neither a measure of cardiac sympathetic tone nor of baroreflex sensitivity], we do not interpret LFnu as a variable of sympathetic activity. However, it can be interpreted as the inverse of parasympathetic activity since a higher value of HFnu will result in a lower value of LFnu. We have added an explanation regarding this fact in the discussion section as different interpretations of LF can be possible among readers.

Those lines are not describing SS and S/PS. Moreover, this description should be added to the “Methods” section.

We have specified further calculations and explanations for the use of these variables in lines 120-122 in the “methods” section as suggested.

Lines 229-234: The authors say “…the influence of temperature on the organism must have triggered excitability of the ANS, causing an increase in blood flow… “. By “increase in blood flow” you mean the cardiac output? Is not vasodilation the effect expected from heat increase?

We did not mean cardiac output, we meant the increase of blood flow due to the widening of blood vessels (vasodilation). Yes, cardiac output is going to increase due to this fact, but it was not the fact we were focusing on in the paragraph.

I would like to personally thank the work of reviewer 3 as it is always interesting to discuss technical aspects with reviewers who have experience in the subject, in this case HRV. In fact, two years ago, during the defense of my doctoral thesis, a member of the tribunal asked me exactly the same questions about the management and interpretation of the LF data.

Thank you very much

Reviewer 3 Report

All comments have been satisfactorily addressed. I have no further comments. 

Author Response

Thank you very much